# A Novel Homozygous Variant in DYSF Gene Is Associated with Autosomal Recessive Limb Girdle Muscular Dystrophy R2/2B

**DOI:** 10.3390/ijms23168932

**Published:** 2022-08-11

**Authors:** Patrizia Spadafora, Antonio Qualtieri, Francesca Cavalcanti, Gemma Di Palma, Olivier Gallo, Selene De Benedittis, Annamaria Cerantonio, Luigi Citrigno

**Affiliations:** 1Institute for Biomedical Research and Innovation, National Research Council, 87100 CS Mangone, Italy; 2Department of Medical and Surgical Sciences, Magna Graecia University, 88100 CZ Germaneto, Italy; 3Department of Experimental and Clinical Medicine, Magna Graecia University, 88100 CZ Germaneto, Italy

**Keywords:** DYSF, LGMDR2/2B, dysferlin, dysferlinopathies, Limb Girdle Muscular Dystrophy

## Abstract

Mutations in the DYSF gene, encoding dysferlin, are responsible for Limb Girdle Muscular Dystrophy type R2/2B (LGMDR2/2B), Miyoshi myopathy (MM), and Distal Myopathy with Anterior Tibialis onset (MDAT). The size of the gene and the reported inter and intra familial phenotypic variability make early diagnosis difficult. Genetic analysis was conducted using Next Gene Sequencing (NGS), with a panel of 40 Muscular Dystrophies associated genes we designed. In the present study, we report a new missense variant c.5033G>A, p.Cys1678Tyr (NM_003494) in the exon 45 of DYSF gene related to Limb Girdle Muscular Dystrophy type R2/2B in a 57-year-old patient affected with LGMD from a consanguineous family of south Italy. Both healthy parents carried this variant in heterozygosity. Genetic analysis extended to two moderately affected sisters of the proband, showed the presence of the variant c.5033G>A in both in homozygosity. These data indicate a probable pathological role of the variant c.5033G>A never reported before in the onset of LGMDR2/2B, pointing at the NGS as powerful tool for identifying LGMD subtypes. Moreover, the collection and the networking of genetic data will increase power of genetic-molecular investigation, the management of at-risk individuals, the development of new therapeutic targets and a personalized medicine.

## 1. Introduction

Mutations in the gene DYSF encoding dysferlin (OMIM#603009) are responsible for a group of autosomal recessive muscular dystrophies known as dysferlinopathies with onset commonly between the second and third decade of life [1]. The Limb Girdle Muscular Dystrophy type R2/2B and the Miyoshi myopathy are the two most frequent clinical presentations. The two forms differ clinically due to the predominant involvement of the proximal pelvic muscles in the LGMDR2/2B and the distal ones of the legs in the Miyoshi myopathy, specifically the gastrocnemius and soleus muscle. Distal myopathy is also known with anterior tibialis onset [2]. In Italy, Limb Girdle Muscular Dystrophy type R2/2B represents the second most frequent form of dystrophy after Calpainopathy [3,4]. Limb Girdle Muscular Dystrophy type R2 (LGMDR2) was first classified as LGMD2B, later in 2018 the European Neuromuscular Centre (ENMC) released a report re-defining and re-classifying LGMD disease group. In the new nomenclature the letter R and D indicate the inheritance mode Recessive or Dominant, the number indicates the order of discovery affected protein [5,6]. In this report, we indicate both classifications with the symbol LGMDR2/2B.

In dysferlinopathies, a severe increase in patient’s plasmatic Creatine Kinase (CK) level is very often observed and its comparable to that observed in dystrophinopathies, but unlike the latter, it remains high for a long time over the years. Cardiac involvement is rare, and moderate respiratory problems can be observed in the later stages of the disease [7]. The ability to walk independently is lost about ten years after the onset of the disease [8]. A wide inter and intra familial phenotype variability has been reported [9,10,11,12].

The DYSF gene encompasses 55 exons and extends for 150 kb of genomic DNA on chromosome 2p13 [13,14,15,16]. Dysferlin is a 230 kDa transmembrane protein composed of a short C-terminal extracellular domain, seven highly conserved cytosolic C2 domains (C2A-G), three Fer domains (FerA, FerB and FerI), two DysF domains (DysFN, DysFC), an intracellular cytoplasmic N-terminal domain. It seems that each C2 domain plays a specific role in dysferlin function. Dysferlin is mainly expressed in the skeletal muscle and in the heart, located in the plasma membrane where it seems to play an important role in plasmalemma repair by regulating vesicle fusion interacting with Caveolin 3 (Cav3) and Mitsgumin 53 (MG53), which are components of the membrane repair system [17,18].

The muscle is continuously subjected to mechanical stresses due to contraction causing micro-lesions of the plasma membrane that must be repaired. The influx of calcium through the lesion triggers a series of cascading events that lead to the recruitment of vesicles into the lesion, their fusion and the formation of a patch that will result in repair of the plasmalemma [19].

Dysferlin appears to be involved in the processes of fusion of the vesicles with each other and with the plasmalemma, in the membrane repair process calcium-mediated [20,21].

Mitsgumin 53, a muscle-specific TRIM (tripartite motif) family protein, works upstream of dysferlin for nuclear recruitment of intracellular vesicles at the injury site, regulates membrane budding, exocytosis in muscle cells and modulates recycling of the membrane interacting with Cav3 [22,23]. Overexpression of MG53 increases dysferlin and caveolin3 levels and their retention at the site of damage muscle membrane [24].

It has been reported that alterations of Cav3, a protein expressed primarily in skeletal and cardiac muscle, results in its retention in the Golgi apparatus, aberrant localization of MG53 and dysferlin, and subsequent defective skeletal muscle membrane repair [25].

Co-immunoprecipitation (Co-IP) tests showed that MG53, dysferlin and Cav3 can form a protein complex by physically interacting with each other when co-expressed in C2C12 myoblast cells [18]. The molecular complex formed by MG53, dysferlin and Cav3 is essential for the repair of muscle membrane damage [26,27].

Furthermore, in recent years an important role of Annexins A1-A6 has been recognized in the repair of the plasma membrane. They interact with membrane phospholipids by relating cytosolic calcium change signals to membrane changes. The Annexins accumulate in the damaged site [28,29], promote blebbing and shedding of the membrane [30,31], aggregation and fusion of the membranes, increasing the closure of the membrane in a calcium-dependent manner [32].

In particular, Annexin 2 (AnxA2) has a synergistic action with dysferlin in the repair process of damaged skeletal myofibre. AnxA2 interacts with the N-terminal C2 domain of the dysferlin [29].

So far, more than 500 different allelic variants have been identified spread throughout the DYSF gene with no hot spot mutational making necessary to analyze the entire gene sequence with very laborious, long and expensive investigations [33]. The recent introduction of more sophisticated molecular methods such as NGS with custom panel, has greatly reduced the costs and time for diagnosis as all genes known related to Limb Girdle Muscular Dystrophy can be analyzed at once [34].

However, a large number of affected patients remain undiagnosed, probably due to the presence of causative variants in genes not yet identified associated with the disease [35].

In the present study, we have conducted the molecular analysis of a genetically unsolved case with clinical suspicion of LGMD and her parents from a consanguineous family of south Italy.

The genetic analysis was conducted by NGS with a custom panel of 40 Muscular Dystrophies associated genes.

We identified in the proband a new missense variant c.5033G>A, p.Cys1678Tyr in the exon 45 of DYSF gene in homozygosity. No mutation was identified in the gene SGCB encoding β-sarcoglycan protein, although an atypical reduction in this protein on the proband’s muscle biopsy was reported.

Consanguineous parents are healthy carriers of the same variant.

The missense variant c.5033G>A, p.Cys1678Tyr is related to Limb Girdle Muscular Dystrophy type R2/2B in the proband and in two other affected members of the family. This variant c.5033G>A has not been previously reported.

## 2. Results

### 2.1. Targeted Next Generation Sequencing

The NGS allowed the simultaneous sequencing of all the exon regions and the exon intron boundaries of the 40 genes associated with LGMD with forty bp padding relative to the splice site.

The 40 targeted-genes panel covers 232.8 kb, 828 amplicons with a size range of 125–375 bp, distributed over two pools (Pool1 and Pool2) of 418 and 410 amplicons, respectively. Gene coverage on the panel was 99.4% and the depth obtained in the run was 350 X.

The data obtained by NGS of the proband were filtered for nonsynonymous variants, variants localized in exons and splice sites, autosomal recessive inheritance, in order to identify variants potentially responsible for LGMDR2/2B, Table 1.

All the variants reported in Table 1, except the two indicated in bold (c.3184_3185insAGGCGG, c.5033G>A in the exon 30 and 45 of DYSF gene, respectively), showed a Minor Allele Frequency (MAF) > 5% and reported as benign, likely benign, tolerated in Mutation taster, Sift, Polyphen bioinformatic software.

The variation c.3184_3185insAGGCGG, p.Q1062delinsQAE (NM_001130455) in the exon 30 of DYSF gene has been reported as polymorphic in the Italian population [36].

The variant c.5033G>A, p.Cys1678Tyr (NM_003494) located in the exon 45 of the DYSF gene was not listed in ClinVAR or Human Gene Mutation Database (HGMD). The effects of this variant on protein function predicted by in silico analysis, suggesting a likely pathogenic role.

The analysis in NGS was extended on the proband’s parents. The comparative analysis of the familial trios allowed the identification of variant c.5033G>A, p.Cys1678Tyr (NM_003494) as a probable cause of the pathological phenotype.

### 2.2. Family Mutational Screening by Sanger Sequencing

The members of the studied family are shown in Figure 1.

Molecular analysis of the proband conducted by Sanger sequencing reconfirmed the missense variant c.5033G>A, p.Cys1678Tyr in homozygosis in the exon45 of the gene DYSF (Figure 2a). Both healthy parents (Figure 2b,c) were heterozygous for this variant.

The genetic investigation was extended to the two sisters moderately affected of the proband (Figure 2d,e). They showed the same missense variant c.5033G>A in homozygosity.

The segregation of the missense variant c.5033G>A, p.Cys1678Tyr in the family was in agreement both with the autosomal recessive transmission of the pathology and with the reported family consanguinity (Figure 2 and Appendix A).

The proband showed also a homozygous 6-bp insertion (AGGCGG) in the exon 30 of DYSF gene causing an addition of two amino acid AE (1062 aa), Table 1. This variant was previously reported in 15% of the Italian population, and so considered as polymorphism [36]. Both parents showed this polymorphism in heterozygous.

No nucleotide changes were identified in the gene SGCB encoding the β-sarcoglycan reduced on muscle biopsy of the proband.

### 2.3. Functional Prediction of the Variant c.5033G>A

We used ten different algorithms to predict effect of the new identified variant c.5033G>A on protein function. All in silico analyzes gave a score associated with a probable deleterious effect of the Cys1678Tyr amino acid substitution in exon 45 of the DYSF gene, Table 2.

### 2.4. Multi-Species Alignment

The nonsynonymus variant identified by NGS determines the replacement of a Cysteine in position 1678 with a Tyrosine. Cysteine 1678 is a highly conserved amino acid in evolution, as shown in the multispecies alignment. Furthermore, the Genomic Evolutionary Rate Profiling (GERP) score 5.13 for the variant c.5033G>A, p.Cys1678Tyr (NM_003494), is indicative of evolutionary constraint, Figure 3.

### 2.5. Dysferlin C2E Model

The predictive models from AlphaFold [37,38] have accuracy competitive with respect to the experimental one and therefore they are very useful for proteins or parts of them for which no experimental models has yet been developed. This is the case of dysferlin because at time there are available experimental structures (PDB files) only covering the range of aminoacid (aa) sequence 1–124 (domain C2A) and 942–1052 (domain inner DysF) while there is not any experimental structure for the domain C2E. The AlphaFold Dysferlin 3D model covers the entire protein sequence with very high confidence score for a large portion of the structure, comprising the domain C2E of interest. In this model, the cysteine 1678 falls in a loop region with very high confidence (Figure 4). This residue is linked by H-bond to aspartic acid 1788 located on the adjacent beta sheet.

When we introduced the aminoacid change Cys > Tyr by using the mutate tool of SPDviewer, the new residue does not showed any type of H bond with adjacent atoms (Figure 5).

The substitution aside, it does not change the electrostatic potential and the hydrophobic patches adjacent but does dramatically change the total local energy (E = −74,897.672e KJ/mol wild type; E = 832,851.313 KJ/mole mutated), Table 3.

## 3. Discussion

Dysferlinopathy includes an autosomal recessively inherited group of Muscular Dystrophies caused by mutations in the DYSF gene, located on chromosome 2p13, codifying dysferlin protein [14].

The exact incidence of dysferlinopathies in the world is unknown, however it is estimated that they represent more than 30% of recessively transmitted Muscular Dystrophies [39].

Dysferlin localizes to the plasma membrane of skeletal muscle and might have a role in the membrane repair machinery in skeletal muscle [40].

In C-elegans, an accumulation of vesicles at the damage site has been reported in the presence of mutation in fer-1 gene which is a homologue of the DYSF gene in mammals.

Likewise, the skeletal muscle cells of dysferlin-null mice, observed by electron microscopy, showed a vesicle accumulation under the damaged plasma membrane [41].

Interestingly, accumulation of small vesicles at sites of muscle damage has been observed in muscle biopsies of patients with dysferlin mutations where the necrotic fibers showed increased numbers of membrane vesicles as compared to the non-necrotic fibers [42].

The enrichment of dysferlin and vesicles at the site of plasma membrane damage observed in muscle cells from patients with mutation in DYSF gene, suggests that dysferlin may be necessary for the fusion of the vesicles with each other and with the plasma membrane for its repair [20].

The repair of the plasma membrane, in fact, appears to take place by intracellular membranes targeted to the damage site in the form of vesicles. The vesicles fuse with each other forming a patch in the membrane repair process. This process takes place in the presence of high levels of calcium due to its intake from the extracellular environment through the injury site [43].

The muscle tissue is a highly specialized tissue that must continuously adapt to the functional demands of the body by modulating mass and contractile properties. The muscle fibers contracting repeatedly, are subjected to mechanical stress and so more susceptible to micro-injuries. For this reason, the plasma membrane repair system must be very sophisticated and efficient [41,44].

Although the exact repair mechanism remains unknown, numerous studies suggest that dysferlin has several protein partners such as Caveolin3, Mitsugumin53, Annexins A1 and A2 in the membrane repair process [18].

Mitsgumin 53 recruits dysferlin containing vesicles at the damage site [45]. The Annexins that bind membrane phospholipids, are instead essential for the process of aggregation and fusion of the vesicles mediated by dysferlin [46,47]. Finally, dysferlin interacts with caveolin3, essential for the formation of caveolae. An accumulation of dysferlin at the injury site has been reported in patients with mutation in the CAV3 gene [48].

In addition, recent in vitro and in vivo studies demonstrated an active role of AnxA2 in the adipogenic conversion of muscle affected by dysferlinopathy [49].

In Italy, the Limb Girdle Muscular Dystrophy type R2/2B is the most frequent clinical presentation of dysferlinopathy second to Calpainopathy [3].

More than 500 different mutations have been identified in the DYSF gene responsible of LGMDR2/2B. The large size of the gene, the lack of mutational hot spots and the great phenotypic variability make early diagnosis difficult [35].

In recent years, high-throughput technologies such as Next-Generation Sequencing have reduced diagnosis times and costs, increasing the identification of new cases of LGMD by 30%, compared to the previous gene-to-gene approach [50]. In fact, the use of a customized NGS panel covering all genes associated with LGMD allows their simultaneous sequencing. Moreover, the high depth of the coverage ensures greater sensitivity for the identification of low frequency variants and low level of mosaicism [51]. However, with custom NGS panel approach, large exonic deletions, long repetitive sequences, causal variants falling into the non-coding portions or in unknown genes, may not be detected [52].

To overcome this limit, if the use of a customized NGS panel did not allow a conclusive diagnosis, Whole Exome Sequencing (WES) or Whole Genome Sequencing (WGS) can be performed [50,53].

The disadvantage in using WES or WGS is to acquire a large number of gene variants and the need to carefully analyze them to identify causative variants of the pathology.

It is important to filter the data properly according to the expected autosomal dominant or recessive inheritance, family history, effect of mutation on protein function, clinical phenotype, and belonging ethnic group.

The use of custom panels remains the cheapest and fastest system for the diagnostic screening of patients with LGMD [54].

Furthermore, segregation analysis of familial trios improves the identification of potentially causative variants of the disease [53].

Interestingly, high throughput sequencing allowed the identification of an increasing number of patients with pathogenic variants in more than one gene associated or not with LGMD. It suggested that the complex phenotype observed in LGMD and the inter and intra-family variability could be due to the synergistic action of primary causative mutations and additive effect of a series of co-causative variants capable of influencing the clinical phenotype [55].

Digenic inheritance has been reported in a subtype of Facioscapulohumeral Muscular Dystrophy [56,57], Congenital Myasthenic Syndrome [58] and in Calpainopathy [59].

In the present study, we report a new missense variant c.5033G>A, p.Cys1678Tyr in the exon 45 of DYSF gene in a consanguineous Italian family (Calabria). This variant potentially causative of LGMDR2/2B has been identified using a custom NGS panel covering 40 LGMD associated genes. The proband and two affected sisters showed the variation in homozygosity. The proband with onset of the disease at age 21 had a severe clinical course with loss of walking ability, while the other two sisters showed only an increase in CPK and weakness in the limb girdle pelvic muscles in climbing stairs. This is in line with inter and intra familial phenotypic variability previously reported, associated to the same mutation in DYSF gene [60]. This is also known for other LGMDs [61].

The consanguineous parents are both heterozygous carriers of the variant c.5033G>A, underlining the autosomal recessive inheritance of dysferlinopathy.

To our knowledge, the missense variant c.5033G>A has never been described in the literature.

This variation determines the replacement of a Cysteine in position 1678 with a Tyrosine. The 1678 Cysteine is an amino acid evolutionary highly conserved falling into one of the seven C2 domains of the protein, probably involved in binding with proteins of the plasma membrane repair machinery in a calcium dependent manner. MutPred2, PhD-SNP, PolyPhen2, CADD, SNAP2, Pmut, Panther, SIFT, PROVEN and Mutation Taster bioinformatic software gave a score associated with a probable deleterious effect of the variant c.5033G>A, p.Cys1678Tyr on protein function, Table 2.

The multispecies alignment obtained by UCSC Genome Browser and the GERP score of 5.13 is indicative of evolutionary constraint of the identified variant (Figure 3).

In support to the probable deleterious effect that amino acid substitution could have on protein activity are the results that we obtained by 3D analysis of the predicted AlphaFold protein model (Figure 5). In particular, the completed dysferlin model allowed us to define that the amino acid variation at position 1678 induced in silico, determines both a dramatic alteration of total local free energy and the breaking of the H- bond between Cys and Arg 1788, in a loop region of the C2E domain that is known to govern the stability of the dysferlin dimer [62].

Moreover, previous studies have identified a different homozygous variation in the same amino acid position as responsible for dystrophinopathy in an Italian family, emphasizing the importance of the region [60].

No nucleotide changes were identified in the SGCB gene encoding β-sarcoglycan reduced on muscle biopsy of the proband.

Secondary reduction in beta sarcoglycan was previously reported in two cases with dysferlinopathy [42]. A secondary dysferlin reduction has been reported in primary caveolinopathy [48] and calpainopathy [63].

The identification of new missense variant c.5033G>A, p.Cys1678Tyr associated with LGMDR2/2B expands the mutational spectrum of DYSF gene.

A defined genetic diagnosis allows a better patient management such as planning cardiological and respiratory follow-ups, physical rehabilitation, non-administration of steroids or immunosuppressive drugs and discourage patients from strenuous physical activity.

Moreover, the identification of new patients with LGMDR2/2B will allow their enrollment in specific clinical trials essential for testing the efficacy and safety of new drugs.

## 4. Materials and Methods

### 4.1. Patients

The 57-year-old proband showed the onset of the disease at 21 years with frequent falls, slowly worsening weakness of the proximal muscles predominantly lower limbs that appear hypotrophic.

The serum creatine phosphokinase (CPK) was higher than 3000 IU/l. Cardiology and respiratory tests were normal.

The electromyography performed on the left vastus lateral showed a myopathic picture.

At 41, she is no longer able to climb stairs or get up from her chair. Moreover, she shows weakness of the deltoids, pectorals, brachial biceps, adductors of the thighs, iliopsoas, quadriceps, anterolateral leg muscles. Hypotrophy of the quadriceps.

The family consisted of seven members (Figure 1), five sisters two of whom III,5 e III,9 (55 and 47 years old, respectively), in addition to the proband (III,1), showed signs of weakness in the limb girdle pelvic muscles in climbing stairs, tendency to fall easily and an increase in CPK. The other two sister III,3 e III,7 (aged 56 and 52) showed no pathological clinical signs. Consanguinity has been reported in the family, the parents are second cousins.

### 4.2. Muscle Biopsy

Biopsy performed on the quadriceps muscle of the proband, revealed marked variability in fiber diameter, centralized nuclei, degeneration and regeneration, foci of sarcoclastosis, many fibers with fissures, some ragged red fibers, increase in interstitial connective tissue. Immunohistochemical examination showed normal signal strength for dystrophin and dystrophin complex proteins, but reduced intensity to β-sarcoglycan.

The data was acquired from the reports shown by the patient. Biopsy analysis was performed following the patient’s hospitalization. The diagnosis was of Limb Girdle Muscular Dystrophy from probable sarcoglycanopathy.

### 4.3. NGS Data Analysis

The molecular-genetic investigation was conducted at the institute for Biomedical Research and Innovation (IRIB), National Research Council CNR, Mangone (CS), Italy. All subjects gave informed consent. Genomic DNA was extracted from peripheral blood lymphocytes of all family’s affected subjects using standard protocol.

We developed a custom panel comprising 40 genes involved in Muscular Dystrophies. The genomics libraries were made using the kit Ion Ampliseq Library preparation kit 2.0. The protocol to form the libraries started from 100 ng input of genomic DNA. Amplicons were partially digested, barcoded, purified and quantified using Qubit™ Fluorometer. The sequencing process conducted on Ion Torrent PGM (Thermo Fisher Scientific-Applied Biosystems, Foster City, CA, USA) platform. The alignment to the human reference genome of GRCh37 (h19) and the variant calling was performed using Torrent Suite v.5.10 (Thermo Fisher Scientific- Applied Biosystems, Foster City, CA, USA) and WANNOVAR for the annotation step (https://wannovar.wglab.org/index.php; accessed on 28 November 2018). The analysis consists of three different levels based on the quality of samples after the alignment (first-level analysis), the annotation and the primary filtering (second-level analysis) and variant detection (third level analysis).

The obtained data were reduced and filtered firstly excluding variants having a minor allele frequency (MAF) > 0.01; next removing synonymous variants with priority those within coding regions; in silico evaluation for identification of potential damaging variants using tools as SIFT and PROVEAN.

The interpretation of the variants was conducted according to the guidelines released by the American College of Medical Genetics (ACMG) [64,65].

### 4.4. PCR and Sanger Sequencing

The variant c.5033G>A located in the exon 45 of the DYSF gene was confirmed by Sanger direct sequencing on an ABI PRISM 3130xl Genetic Analyzer (Thermo Fisher Scientific-Applied Biosystems, Foster City, CA, USA).

Polymerase Chain Reaction (PCR) amplification of the exon 45 of the DYSF gene was performed with the following primers: Forward 5′-GGGTGCCCTGTGTTGGCTGAC-3′; Reverse 5′-GCAGGCAGCCAGCCCCCATC-3′. The reaction was performed in a 25 mL volume containing 10 ng of genomic DNA, 1,25 U of AmpliTaq Gold DNA Polymerase (Applied Biosystem) and standard reagents according to manufacturer’s instruction. PCR profile includes an initial denaturation step of 10 min at 95 °C pre-denaturation to activate DNA Polymerase followed by 35 cycles of 94 °C 15 s denaturation, 67 °C 45 s annealing, 72 °C 45 s extension and a final elongation of 10 min at 72 °C.

The PCR reaction was performed in a ProFlexPCR System (Thermo Fisher Scientific- Applied Biosystems, Foster City, CA, USA).

The amplified PCR products were confirmed by agarose gel electrophoresis, purified by Centri-Sep Spin Columns (Applied Biosystem-Thermo Fisher Scientific, Foster City, USA), extended using BigDye 3.1 termination ready reaction mix (Applied Biosystem-Thermo Fisher Scientific, Foster City, USA) and sequenced by ABI PRISM 3130xl Genetic Analyzer.

All the variations of interest identify by NGS were confirmed by direct resequencing.

### 4.5. Bioinformatic Software

Ten different algorithms were used to predict the functional analysis of the variant c.5033G>A, p.Cys1678Tyr in DYSF gene: MutPred2, PhD-SNP, PolyPhen2, CADD, SNAP2, Pmut, Panther, SIFT, PROVEN and Mutation Taster. The UniProt database was used to obtain the amino acid sequence of the human Dysferlin protein used in this investigation (UniProt ID: O75923). Moreover, we used PolyPhen2, Mutation Taster and UCSC Genome Browser to test the multi-species alignment. The links for each software package are shown in Table 2.

### 4.6. In Silico 3D Structure Analysis

In order to analyse the dysferlin 3D structure, we used 3D model data from AlphaFold that is an Artificial Intelligence system developed by DeepMind capable of accurately predicting a protein’s 3D structure from its amino acid sequence [27,28]. We used the PDBviewer software, version 4.1, to view and analyze the 3D structure codified into the PDB file AF-O75923-F1-model_v1 for both wild type and mutated dysferlin. The Alpha Fold database is available online at https://alphafold.ebi.ac.uk/ (accessed on 30 June 2022), selectable also from the Uniprot database at https://www.uniprot.org/ (accessed on 30 June 2022).

## 5. Conclusions

Our study reveals that the new missense variant c.5033G>A, p.Cys1678Tyr (NM_003494), located in the exon 45 of DYSF gene, is related to Limb Girdle Muscular Dystrophy type R2/2B in a patient with clinical suspicion of LGMD from a consanguineous family of south Italy.

The variant c.5033G>A, p.Cys1678Tyr segregates with the pathological phenotype of two other affected members of the family and it is agrees with the autosomal recessive inheritance of LGMDR2/2B.

This variation determines the replacement of a Cysteine in position 1678 with a Tyrosine and falls into a highly conserved evolutionary domain as showed by the multispecies alignment and the Genomic Evolutionary Rate Profiling positive score.

In support of the probable deleterious effect of the variant on protein activity are the scores obtained by in silico analysis using MutPred2, PhD-SNP, PolyPhen2, CADD, SNAP2, Pmut, Panther, SIFT, PROVEN and Mutation Taster and the 3D analysis of the predicted AlphaFold protein model.

The identification of new missense variant c.5033G>A, p.Cys1678Tyr associated with LGMDR2/2B expands the mutational spectrum of DYSF gene pointing the NGS as powerful tool for identifying LGMD subtypes.

The identification of multiple families affected by LGMDR2/2B could help us in the identification of additional genetic or epigenetic factors contributing to phenotypic variability observed in LGMDR2/2B and in identifying new potential candidates in translational medicine as therapeutic targets.

Finally, an accurate and early characterization of the different LGMD subtypes gives patients the opportunity to participate in clinical trials, essential for the identification of new therapies still not available for most LGMD forms.

## Figures and Tables

**Figure 1 ijms-23-08932-f001:**
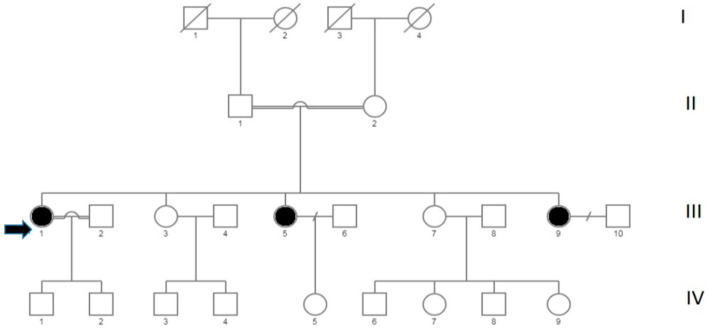
Pedigree of the family with LGMD. Solid symbols represent the affected state, open symbols represent the unaffected individuals. Double lines indicate consanguineous marriage. The proband is indicated with an arrow.

**Figure 2 ijms-23-08932-f002:**
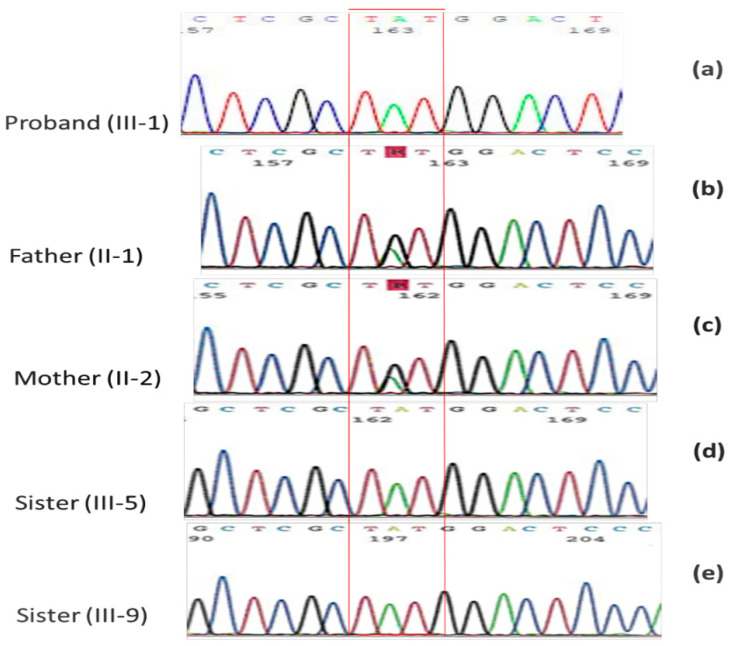
Electropherogram obtained from Sanger sequencing. (**a**) homozygous missense variation in DYSF c.5033G>A, p.Cys1678Tyr (NM_003494) in the proband; (**b**,**c**) the healthy father and mother heterozygous for c.5033G>A p.Cys1678Tyr variation; (**d**,**e**) the proband’s sisters homozygous for the same missense variation.

**Figure 3 ijms-23-08932-f003:**
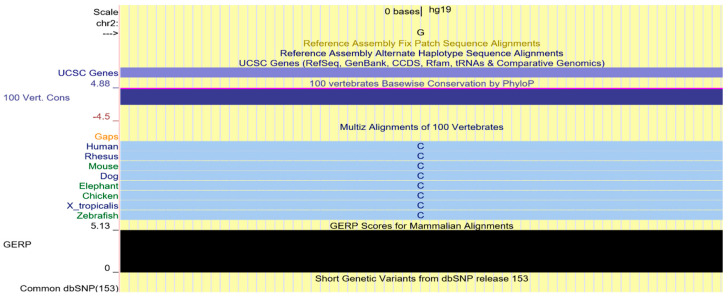
Multi-species alignment obtained by UCSC Genome Browser and the GERP score.

**Figure 4 ijms-23-08932-f004:**
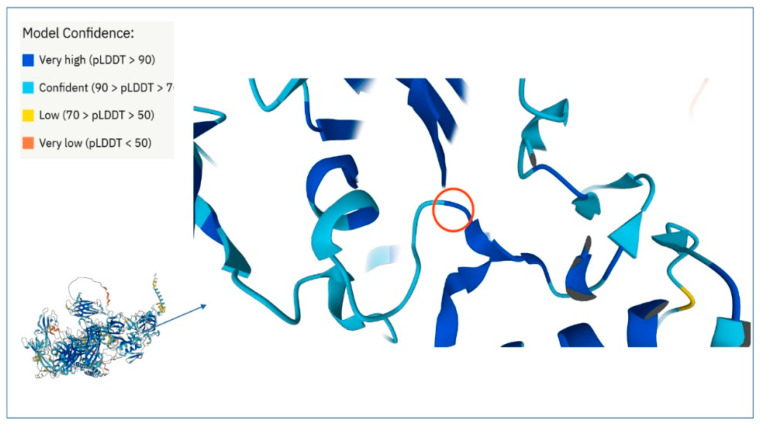
3D dysferlin image from AlphaFold. In lower left angle there is the image of the whole protein structure and zoomed the protein portion showing the loop containing the 1678 residue indicated by the red circle. The AlphaFold model confidence score is indicated by the color code in the upper left table.

**Figure 5 ijms-23-08932-f005:**
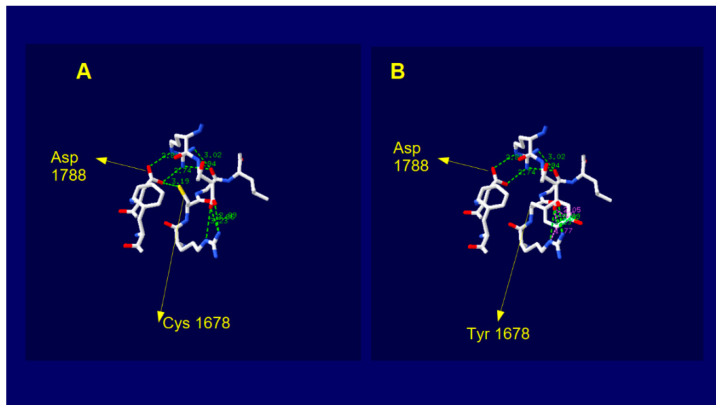
3D structure images of the AlphaFold dysferlin model showing the atomic interactions centered on the residue 1678 and in a range of 8Angstrom. (**A**) wild type model with cysteine residue 1678 forming a H-bond with Aspartic at position 1788; (**B**) model upon Cysteine > Tyrosine change. Green dashed lines and green numbers indicate the H-bond and the respective interatomic distances, respectively. Residues 1678 and 1788 are indicated.

**Table 1 ijms-23-08932-t001:** Variations identify by NGS after filtering. In bold are indicated the two variants investigated in the family trios.

Chr	WT	nt Change	Func. Refgenze	Gene	ExonicFunc. Refgene	AAChange.Refgene	dbSNP	Inher
chr1	T	C	exonic	POMGNT1	nonsynonymous SNV	POMGNT1:NM_001290129:exon20:c.A1801G:p.M601V	rs6659553	hom
**chr2**	**-**	**AGGCGG**	**exonic**	**DYSF**	**nonframeshift insertion**	**DYSF:NM_001130455:exon30:c.3184_3185insAGGCGG:** **p.Q1062delinsQAE**	**rs398123779**	**hom**
**chr2**	**G**	**A**	**exonic**	**DYSF**	**nonsynonymous SNV**	**DYSF:NM_003494:exon45:c.G5033A:p.C1678Y**	**.**	**hom**
chr3	C	G	exonic	DAG1	nonsynonymous SNV	DAG1:NM_001177639:exon2:c.C41G:p.S14WW	rs2131107	hom
chr3	T	C	exonic	GMPPB	nonsynonymous SNV	GMPPB:NM_013334:exon5:c.A551G:p.Q184R	rs1466685	hom
chr5	A	C	exonic	MYOT	nonsynonymous SNV	MYOT:NM_006790:exon2:c.A220C:p.K74Q	rs6890689	hom
chr17	A	G	exonic	GAA	nonsynonymous SNV	GAA:NM_000152:exon3:c.A596G:p.H199R	rs1042393	hom
chr17	G	A	exonic	GAA	nonsynonymous SNV	GAA:NM_000152:exon3:c.G668A:p.R223H	rs1042395	hom
chr17	G	A	exonic	GAA	nonsynonymous SNV	GAA:NM_000152:exon17:c.G2338A:p.V780I	rs1126690	hom
chrX	C	T	exonic	DMD	nonsynonymous SNV	DMD:NM_004014:exon5:c.G623A:p.R208Q	rs1800280	hom

**Table 2 ijms-23-08932-t002:** Functional prediction of the missense mutation in DYSF gene (c.5033G>A, p.Cys1678Tyr).

Tool	Effect	Score/ Reliability Index	Link
MutPred2	Deleterious	0.90	http://mutpred.mutdb.org/ (accessed on 21 July 2021)
PhD-SNP	Deleterious	10	https://snps.biofold.org/phd-snp/phd-snp.html (accessed on 21 July 2021)
PolyPhen2	Deleterious	1	http://genetics.bwh.harvard.edu/pph2/ (accessed on 21 July 2021)
CADD	Deleterious	4.19 (PHRED 29.5)	https://cadd.gs.washington.edu/ (accessed on 21 July 2021)
SNAP2	Deleterious	66	https://www.rostlab.org/services/snap/ (accessed on 21 July 2021)
Pmut	Deleterious	0.87	https://wwwmmb.irbbarcelona.org/PMut/ (accessed on 21 July 2021)
Panther	Deleterious	0.86	http://www.pantherdb.org/tools/ (accessed on 21 July 2021)
SIFT	Deleterious	0.04	https://sift.bii.a-star.edu.sg/ (accessed on 21 July 2021)
PROVEN	Deleterious	−10.41	http://provean.jcvi.org/index.php (accessed on 21 July 2021)
Mutation Taster	Deleterious	0.99	https://www.mutationtaster.org/ (accessed on 21 July 2021)

**Table 3 ijms-23-08932-t003:** Energy values (KJ/mole) of Wild type (Wt) and mutated dysferlin.

	Bonds	Angles	Torsion	Improper	NonBonded	Electrostatic	Constraint//	TOTAL
**Wt dysferlin**	16,540.810	18,356.452	14,731.280	1306.953	−65,364.79	−60,468.37	0.0000//	E = −74,897.672
**Mutated**	16,542.059	18,368.218	14,725.421	1308.030	842,455.48	−60,547.98	0.0000//	E = 832,851.313

## Data Availability

All datasets generated or analysed during the study are available upon request.

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
