# Peer review of "A Novel Homozygous Variant in DYSF Gene Is Associated with Autosomal Recessive Limb Girdle Muscular Dystrophy R2/2B"

_ijms, 2022, doi:10.3390/ijms23168932_

Round 1

Reviewer 1 Report

This manuscript indicated novel mutation in DYSF gene, which affects LGMDR2/2B phenotypes. This funding of the mutation is very important. But some corrections may be needed. It is better to add the functional analysis of the mutation, causing the disease conditions. For example, functional differences of the mutant protein with amino acid substitution. In addition, it is better to add haplotype analysis of the mutation. Also, it is better to show principal component analysis of molecular descriptors of chemical structure in the mutant protein.

Author Response

Point 1: This manuscript indicated novel mutation in DYSF gene, which affects LGMDR2/2B phenotypes. This funding of the mutation is very important. But some corrections may be needed. It is better to add the functional analysis of the mutation, causing the disease conditions. For example, functional differences of the mutant protein with amino acid substitution.

Response 1: We agree with the Reviewer that it would be important to add functional analysis of the disease-causing mutation in the work, for this reason we used ten different algorithms, MutPred2, PhD-SNP, PolyPhen2, CADD, SNAP2, Pmut, Panther, SIFT, PROVEN and Mutation Taster to predict effect of the new identified variant c.5033G>A, p.Cys1678Tyr (NM_003494) on protein function. All in silico analyzes conducted, showed a score associated with a probable deleterious effect of the Cys1678Tyr amino acid substitution in the protein. In the current version we have included a new sub-paragraph 2.3 and 4.4 in Results and Materials Methods respectively. All the scores obtained for the variant Results sectionc.5033G>A are shown in Table 2. The used algorithms have been reported in the literature by Pereira GRC et al. 2020 (PLoS One. 2; 15(3):e0229730. doi: 10.1371/journal.pone.0229730) and Montenegro LR et al.2021 (Clinics (Sao Paulo). 22; 76:e2052. doi: 10.6061/clinics/2021/e2052) as important tools for studying the effects of the genetic mutation on protein function.

We have also reported in the Results section, subparagraph 2.4, Figure 3, the Multi-species alignment obtained by UCSC Genome Browser and the GERP score, indicative of the evolutionary constraint of the amino acid residue.

Point 2: In addition, it is better to add haplotype analysis of the mutation.

Response 2: We thank the Reviewer for the suggestion that would be better to add the haplotype analysis of the mutation. The analysis conducted by NGS using a custom panel comprising 40 genes (all exonic portions and splicing sites) associated with LGMD allowed the simultaneous comparison of data in the familial trios. The variant c.5033G>A, p.Cys1678Tyr (NM_003494), segregates with the pathological phenotype in the family in which, in addition to the proband and the parents, two other affected members of the family were analyzed Results section, subparagraph 2.2. The variant is also perfectly in agreement with the autosomal recessive transmission of LGBDR2 / 2B and with the presence of consanguinity in the family (Figure 1 and Figure 2). The filtered data relating to the proband are included in the Results section, Table 1.

Point 3: Also, it is better to show principal component analysis of molecular descriptors of chemical structure in the mutant protein.

Response 3: We agree with the Reviewer that it would be important to add principal component analysis of molecular descriptors of chemical structure in the mutant protein and we thank him for the suggestion which will have to be the subject of a future study. In fact, the size of the dysferlin makes extremely complex to generate the numerical variables, in a context of Molecular Dinamics Simulations, necessary for the elaboration of the principal components, in such a short time. Dysferlin is a protein consisting of 2080 aa and there are many variables to be generated, that we had not planned to do. Moreover, since all the analyzes performed are predictive of the influence of the variant c.5033G>A identified on protein function, we have replaced all the terms in the text "as responsible" or "causes" with "related to" or "probably responsible" or "associated with" as suggested by the Reviewer 2.

Finally, as indicated by the Reviewer we have improved Introduction, cited References, Research design, Method, Results and Conclusions.

We thank the Referee for the important suggestion.

Reviewer 2 Report

In this manuscript, Spadafora and colleagues identify a new missense mutation in the DYSF gene potentially related to dysferlinopathy, i.e. limb-girdle muscular dystrophy 2B (or R2 following the new denomination). The genetic analysis is straightforward and report of a new missense mutation linked to this disease is important. However, I think the Authors should address several shortcomings to support their claim and improve significance of presented work:

-       The title states that the novel identified mutation “causes” LGMD-R2. This claim is not supported by this work as causality was not established through animal or cell models. Title and claims through the text in that regard should be appropriately toned down to “correlation”, “relation”, “association”, etc...

-       In the abstract, the Authors state that an NGS panel with 40 target genes was used. The Authors should report all details related to the gene panel used, other mutations found (or lack thereof) and precise details about the NGS nature of the gene panel itself (exome sequencing, RNA-sequencing, whole gene sequencing, etc…) in the Results section.

-       The Authors state that muscle biopsies were collected from the proband and the other two homozygous relatives. Have the Authors performed molecular and histological analyses on these biopsies? What are the levels of RNA and protein expression for DYSF in homozygous versus heterozygous (or unrelated) carriers? Histopathological assessment should also be reported for the homozygous carriers.

-       (important, please double-check) the Sanger chromatograms of the heterozygous carrier parents displayed in Figure 2 look exactly the same. It is highly unlikely that this is really the case and likely underscores a chromatogram duplication. Please provide the actual chromatograms of heterozygous carrier parents, as well as for the additional two homozygous carriers.

-       The found missense mutation (c.5036G>A, p.Cys1678Tyr on NM_001130455, i.e. on transcript variant 2 of DYSF gene) is reported as variant of unknown significance in dbSNP (https://www.ncbi.nlm.nih.gov/snp/rs753279446#variant_details ) and ClinVar (https://www.ncbi.nlm.nih.gov/clinvar/variation/288830/?new_evidence=true ) databases. This needs to be acknowledged and reported in the paper, including the reported minor allele frequencies and previous reports or databases reporting this VUS. Beyond the virtual modeling predictions reported by the Authors, are there predictions of pathogenicity score for this variant in widely used tools like SIFT, CADD, PolyPhen2? What is the GERP score for this mutation, i.e. an indirect measure of potential mutation importance based on site resistance to mutation through multi-species alignment?

Author Response

Point 1: The title states that the novel identified mutation “causes” LGMD-R2. This claim is not supported by this work as causality was not established through animal or cell models. Title and claims through the text in that regard should be appropriately toned down to “correlation”, “relation”, “association”, etc...

Response 1: We agree with the Reviewer’s suggestion and we have replaced all the terms "causes" or “responsible” or “pathological role” in the test with “associated with”, “related to” or “probable pathological role”.

Point 2: In the abstract, the Authors state that an NGS panel with 40 target genes was used. The Authors should report all details related to the gene panel used, other mutations found (or lack thereof) and precise details about the NGS nature of the gene panel itself (exome sequencing, RNA-sequencing, whole gene sequencing, etc…) in the Results section.

Response 2: As indicated by the Reviewer, we have reported the details of the panel used by Next Generation Sequencing in the Results section, subparagraph 2.1 and the mutations identified in the proband after filtering, Results section, Table 1.

Point 3: The Authors state that muscle biopsies were collected from the proband and the other two homozygous relatives. Have the Authors performed molecular and histological analyses on these biopsies? What are the levels of RNA and protein expression for DYSF in homozygous versus heterozygous (or unrelated) carriers? Histopathological assessment should also be reported for the homozygous carriers.

Response 3: We agree with the Reviewer that it would be important to measure the level of RNA and protein expression for DYSF in homozygous versus heterozygous carriers, however we did not perform the muscle biopsy. It was performed following the proband’s hospitalitation in another Institution. We have reported in the work all the information contained in the biopsy report provided by the patient. The proband's sisters never performed a biopsy analysis. The guidelines agree that it is advisable at first perform the molecular analysis with the new NGS technologies and only subsequently, if the genetic analysis has given negative results, to perform a biopsy as it is an invasive technique. We have included in Materials and Methods section, in a new sub-section 4.2 for muscle biopsy.

Point 4: (important, please double-check) the Sanger chromatograms of the heterozygous carrier parents displayed in Figure 2 look exactly the same. It is highly unlikely that this is really the case and likely underscores a chromatogram duplication. Please provide the actual chromatograms of heterozygous carrier parents, as well as for the additional two homozygous carriers.

Response 4: In Results section, Figure 2, by mistake, we have displayed the same chromatogram for the proband's parents. In the current version of the manuscript, As indicated by the Reviewer, we have shown the exact chromatograms of the proband’s parents and the two affected sisters carrying the same variant in homozygosity.

Point 5: The found missense mutation (c.5036G>A, p.Cys1678Tyr on NM_001130455, i.e. on transcript variant 2 of DYSF gene)is reported as variant of unknown significance in dbSNP (https://www.ncbi.nlm.nih.gov/snp/rs753279446#variant_details ) and ClinVar (https://www.ncbi.nlm.nih.gov/clinvar/variation/288830/?new_evidence=true ) databases. This needs to be acknowledged and reported in the paper, including the reported minor allele frequencies and previous reports or databases reporting this VUS. Beyond the virtual modeling predictions reported by the Authors, are there predictions of pathogenicity score for this variant in widely used tools like SIFT, CADD, PolyPhen2? What is the GERP score for this mutation, i.e. an indirect measure of potential mutation importance based on site resistance to mutation through multi-species alignment?

Response 5: By mistake, we have reported the variant identified as, c.5036G> A, p.Cys1678Tyr instead of c.5033G> A, p.Cys1678Tyr (NM_003494). No variant of uncertain significance was reported in this position on the transcript NM_003494. We agree with the Reviewer regarding the position c.5036G>A, NM_001130455 which concerns a Cys1679Tyr for which there is a rs753279446. In the current version of the work, we have corrected the exact location of the identified variant in all Sections of the text.

We agree with the reviewer’s suggestion to include the predictions of pathogenicity score for this variant obtained by SIFT, CADD, PolyPhen2. We have included in the Results section a new subsection 2.3 and a new table (Table 2) with the score obtained from ten different algorithms used for the in silico analysis of the pathogenicity prediction of the identified variant c.5033G>A, p.Cys1678Tyr (NM_003494). Moreover, as suggested by the Reviewer, we performed the multispecies alignment using the UCSC Genome Browser and obtained the GERP score. We have included these data in the Results section, subsection 2.4 and Figure 3.

Finally, as suggested by Reviewer, we have improved Introduction, cited References Research design, Methods, Results and Conclusion.

We thank the Referee for the important suggestion.

Round 2

Reviewer 1 Report

This manuscript indicated new mutations of dysferin associated with LGMD. This mutations were very important for understanding of LGMD phenotypes. But, it is better to add advantage(s) and disadvantage(s) in discussion section. 

Author Response

Point 1: This manuscript indicated new mutations of dysferin associated with LGMD. This mutations were very important for understanding of LGMD phenotypes. But, it is better to add advantage(s) and disadvantage(s) in discussion section. 

Response 1: We thank the Reviewer for the suggestion. We have added the advantages and disadvantages in the discussion section as indicated.

We have also improved Introduction, cited References, Research design, Method, Results and Conclusions.

We thank the Referee for the important suggestion.

Reviewer 2 Report

The Authors addressed all points. However, the chromatograms for proband's mother and father still look identical (second and third from the top). The same is true for the proband's sisters' chromoatograms (fourth and fifth from the top). I would like to ask the Authors to review this again and to supply five original chromatogram files (one per subject) together with the revised manuscript. the chromatograms should span from beginning to end of sequenced fragments, i.e. from nucleotide 1 of sequenced PCR fragment, and not only the 20-30nt spanning around the mutation site. This is a central point and image duplication cannot be accepted.

Round 3

Reviewer 1 Report

This manuscript showed important points of the results clearly. 

Reviewer 2 Report

The Authors have addressed my concerns and provided original chromatograms for the sequences. I suggest incorporating those - with the appropriate legend - as supplementary files for the final paper.